# Histological Profiling of the Human Umbilical Cord: A Potential Alternative Cell Source in Tissue Engineering

**DOI:** 10.3390/jpm12040648

**Published:** 2022-04-18

**Authors:** Cristina Blanco-Elices, Jesús Chato-Astrain, Alberto González-González, David Sánchez-Porras, Víctor Carriel, Ricardo Fernández-Valadés, María del Carmen Sánchez-Quevedo, Miguel Alaminos, Ingrid Garzón

**Affiliations:** 1Doctoral Programme in Biomedicine, University of Granada, E18071 Granada, Spain; cblanco@ugr.es; 2Tissue Engineering Group, Department of Histology, University of Granada, E18016 Granada, Spain; jchato@ugr.es (J.C.-A.); david.s.p.94@gmail.com (D.S.-P.); vcarriel@ugr.es (V.C.); rfdezvalades@me.com (R.F.-V.); mcsanchez@ugr.es (M.d.C.S.-Q.); 3Instituto de Investigación Biosanitaria ibs.GRANADA, E18012 Granada, Spain; 4Department of Molecular Biology (IDIVAL), Faculty of Medicine, University of Cantabria, E39011 Santander, Spain; alberto.gonzalezgo@alumnos.unican.es; 5Division of Pediatric Surgery, University Hospital Virgen de las Nieves, E18014 Granada, Spain

**Keywords:** umbilical cord, MSC, vascular differentiation

## Abstract

The embryonic development of the human umbilical cord (hUC) is complex, and different regions can be identified in this structure. The aim of this work is to characterize the hUC at in situ and ex vivo levels to stablish their potential use in vascular regeneration. Human umbilical cords were obtained and histologically prepared for in the situ analysis of four hUC regions (intervascular—IV, perivascular—PV, subaminoblastic—SAM, and Wharton’s jelly—WH), and primary cell cultures of mesenchymal stem cells (hUC-MSC) isolated from each region were obtained. The results confirmed the heterogeneity of the hUC, with the IV and PV zones tending to show the higher in situ expression of several components of the extracellular matrix (collagens, proteoglycans, and glycosaminoglycans), vimentin, and MSC markers (especially CD73), although isolation and ex vivo culture resulted in a homogeneous cell profile. Three vascular markers were positive in situ, especially vWF, followed by CD34 and CD31, and isolation and culture revealed that the region associated with the highest expression of vascular markers was IV, followed by PV. These results confirm the heterogeneity of the hUC and the need for selecting cells from specific regions of the hUC for particular applications in tissue engineering.

## 1. Introduction

The human umbilical cord (hUC) is a conduit about 50–60 cm long and 1–2 cm in diameter, connecting the abdominal wall of the fetus to the placenta for exchange of nutrients and gasses during gestation [1]. The hUC consists of a vein and two arteries surrounded by an embryonic connective mucous tissue, known as Warton’s jelly, that contains a very interesting population of mesenchymal stem cells (MSC) in an abundant extracellular matrix (ECM) [2]. Embryologically, the hUC is formed by the migration of extraembryonic mesoderm cells that connect with the chorionic mesoderm. Then, these structures are compressed into a slender cord covered by amniotic epithelium and vascularized by allantoid vessels [3].

The hUC is a heterogeneous structure in which several cell subpopulations and ECM configurations can be found [4]. Although different classification algorithms have been proposed, most reports consider four main regions in the hUC [1]: the superficial subamnioblastic zone (SAM) that is covered by amnioblast cells, the Wharton zone (WH) allocated between the SAM region and the blood vessels, the perivascular zone (PV) surrounding the vessels, and the intervascular zone (IV) found in the central space of the cord, among the three vessels of the hUC [1,5,6]. Although further studies are needed, some reports demonstrated that each of these regions may differ, both in the histological structure of the ECM, and in the properties of the cells found in each zone [6,7]. The complex embryonic development of the hUC, and the considerable number of structures involved, may explain this heterogeneity. 

Cells isolated from the hUC demonstrated a high expansion potential, plasticity, and immunomodulatory properties, and have been proposed as potentially useful in stem cell therapies [8,9]. In fact, MSC isolated from the hUC have been extensively used in regenerative medicine of the nerve tissue, bone, cartilage, skin, cornea, and oral mucosa, among others [10,11,12,13]. In addition, these cells were shown to be potentially useful in vascular tissue engineering, and it has been demonstrated that MSC isolated from the hUC can be used as a source of cells with endothelial differentiation potential [14]. These cells are identified as immuno-privileged and can be used clinically due to their immunomodulatory properties and the absence of ethical concerns [15]. However, the results obtained with the clinical application of these cells is variable, and most clinical studies lack sufficient scientific evidence to support the routine use of these cells in humans [16]. One of the factors related to this variability could be the structural and histological heterogeneity of the hUC. Therefore, identification and characterization of the different regions found in the hUC could contribute to determining the potential application of each region to specific therapeutic purposes [7], including the generation of bioartificial tissues by tissue engineering and other applications in regenerative medicine [6,17,18].

In the present work, we have carried out a histological, histochemical, and immunohistochemical analysis of the hUC cells and ECM in order to evaluate the profile of each region of the hUC as a source of cells for use in vascular tissue engineering.

## 2. Materials and Methods

### 2.1. In Situ Histological and Histochemical Characterization of the Human Umbilical Cord

With the consent of the parents, five human umbilical cords (hUCs) were obtained from full-term newborns delivered by previously-scheduled cesarean section due to obstetric reasons. Samples were washed in 1× phosphate buffered saline (PBS) and cut into 1.5 cm-length pieces. For histological analysis, transversal sections of the five hUCs were fixed in 4% formaldehyde for 48 h at room temperature, washed, dehydrated, cleared, and embedded in paraffin following standard histological protocols. Tissue sections were obtained, dewaxed, and stained with hematoxylin and eosin (HE) and examined with a Nikon Eclipse 90i light microscope (Nikon Corp., Tokyo, Japan). 

For scanning electron microscopy (SEM), samples were fixed in 2.5% buffered glutaraldehyde in 0.05 M cacodylate buffer at 4 °C overnight and then washed three times in the same buffer at 4 °C. Fixed samples were dehydrated in increasing concentrations of acetone (30%, 50%, 70%, 90%, and 100%) and completely dried using the critical point drying method. All these reagents were obtained from Panreac Química S.L.U., Barcelona, Spain. Dried samples were mounted on aluminum stubs, covered with gold-palladium and examined with a FEI Quanta 200 scanning electron microscope using the high vacuum mode (FEI, Eindhoven, The Netherlands).

Analysis of relevant ECM components of the hUC was performed by histochemistry. First, collagen fibers were identified by using Picrosirius red (PR) methods. Briefly, tissue sections were stained with a Sirius red F3B working solution for 30 min and counterstained with Harris Hematoxylin for 5 min. For proteoglycans detection, alcian blue (AB) was used by staining dewaxed sections in the AB reagent solution for 30 min, followed by PBS washing. To detect ECM glycosaminoglycans, sections were stained with periodic acid-Schiff (PAS). In this case, tissues were incubated in a 0.5% periodic acid solution for 5 min as an oxidant, followed by incubation in Schiff reagent for 15 min and counterstaining with Harry’s hematoxylin for 1 min (all reagents were from PanReac Química S.L.U.). Normal human tissues were used as technical controls for each method.

### 2.2. Generation of Primary Ex Vivo Cultures of Human Umbilical Cord Mesenchymal Stem Cells (hUC-MSC)

For the ex vivo analysis, hUC biopsies were carefully analyzed using a stereo microscope, and biopsies corresponding to each topographical region of the hUC (IV, PV, SAM, and WH) were obtained using a 1mm surgical punch. Once pre-dissected hUC zones were obtained, tissue sections were digested for 6h at 37 °C in a 2 mg/mL solution of *Clostridium histolyticum* type I collagenase (Gibco-Thermo Fisher Scientific, Waltham, MA, USA). Isolated cells were collected by centrifugation and seeded in 75 cm^2^ culture flasks (Sarstedt, Nümbrecht, Germany) using Amniomax-C100 culture medium (Gibco-Thermo Fisher Scientific) supplemented with 10% fetal bovine serum (FBS) and 1% antibiotics/antimycotics (Merck, Burlington, MA, USA). Cells were kept using standard cell culture conditions at 37 °C in a humidified incubator with 5% CO_2_ to generate cultured human umbilical cord mesenchymal stem cells (hUC-MSC). Once the cells reached 70% confluence, they were trypsinized with 0.25% trypsin-ethylenediaminetetraacetic acid -EDTA- (Merck). All hUC-MSC were used at cell passages 1 to 3.

hUC-MSC isolated from each region were analyzed by flow cytometry to determine the expression of typical MSCs markers using a human MSC Analysis BD Stemflow™ kit (BD Biosciences, Franklin Lakes, NJ, USA). In brief, 5 × 10^5^ cells of each zone were placed in flow cytometry tubes and washed with 2mL of staining buffer (R&D Systems Inc., Minneapolis, MN, USA). Then, the Fc receptors were blocked by incubating the cells for 5 min with 2 mL of PBS containing 0.1% bovine serum albumin and 0.1% FBS. Next, cells were stained with a positive cocktail of markers (CD73, CD90, and CD105) and a negative cocktail (CD11b, CD19, CD34, CD45, and HLA-DR) and incubated for 45 min at 4 °C in darkness. Once cells were labeled, the cells were placed in a staining buffer and analyzed using a FACSCalibur flow cytometer (BD Biosciences), and the percentage of positive cells was quantified in each study group.

### 2.3. In Situ Immunohistochemical Characterization of the Human Umbilical Cord

Analysis of relevant stromal and MSC markers (vimentin, CD73, CD90, and CD105), ECM components (collagens type III and IV, fibronectin, and versican), and vascular cell markers (CD31, CD34, and vWF) was performed by immunohistochemistry. As detailed in Appendix A, deparaffinized tissue sections were rehydrated and treated with 0.01 M citrate buffer or chondroitinase for antigen retrieval. Samples were then incubated for 10 min in 3% H_2_O_2_ to quench endogenous peroxidase. Each of the succeeding steps was followed by a thorough rinse in 1× PBS and were performed in a humid chamber to prevent tissue desiccation. Non-specific staining was blocked by incubation in 2.5% horse serum and casein (Vector Laboratories, Burlingame, CA, USA). Then, samples were incubated overnight with primary antibodies at 4 °C. After that, samples were incubated with secondary antibodies at room temperature for 1h. The complex epitope antibody was detected using 3,3-diaminobenzidine -DAB- (Vector Laboratories), and tissue sections were counterstained for 20 s using Harry’s hematoxylin (Gibco-Thermo Fisher Scientific). Human control tissues were used as technical controls. In all cases, images were obtained using a Pannoramic^®^ DESK II DW scanner (3D Histotech, Budapest, Hungary). The antibodies used and technical details of the immunohistochemical procedure are summarized in Appendix A.

### 2.4. Ex Vivo Immunofluorescence Analysis of hUC-MSC

Relevant MSC markers were analyzed by immunofluorescence in cultured hUC-MSC. Briefly, 1 × 10^4^ cells were cultured in chamber slides (Nunc/Thermo Fisher Scientific, Waltham, MA, USA), fixed with 70% ethanol, washed in 1× PBS, and blocked for 30 min with normal horse serum and casein (Vector Laboratories). Then, samples were incubated for 1 h at room temperature with primary antibodies. Subsequently, samples were washed in 1× PBS and incubated for 1 h with FITC-conjugated anti-mouse secondary antibodies (Merck). The technical details of the immunofluorescent procedure are summarized in Appendix A. Samples were finally counterstained with 4′,6-diamidino-2-phenylindole -DAPI- (Vector Laboratories) and analyzed using an Eclipse 90i microscope (Nikon Corp.).

### 2.5. Quantification and Statistical Analysis

In the first place, the results obtained for the histochemical analyses (PR, AB, and PAS) and for the immunohistochemical analysis of ECM components (collagens type III and IV, fibronectin and versican) were quantified as previously described [19]. In brief, the intensity of the histochemical signal was calculated by using the automatic analysis tool of the ImageJ software (National Institutes of Health, Bethesda, MD, USA). In each region of each hUC, we randomly selected six square areas of 384 × 384 pixels, and the average signal intensity was calculated in each area as staining intensity units (I.U.). In the second place, results corresponding to the stromal, MSC, and vascular cell markers (vimentin, CD73, CD90, CD105, VIM, CD31, CD34, and vWF) were quantified by determining the percentage of positive cells in each region of each sample. All analyses were carried out using 15 independent samples (*n* = 15).

Averages and standard deviations were calculated for each sample and each analytical method, and comparison between groups was performed using the Mann–Whitney test, since distributions demonstrated to be non-normally distributed, according to the Shapiro–Wilk test. Values of *p* < 0.05 were considered statistically significant in a two-tailed test. Statistical testing was carried out using the Real Statistics Resource Pack software (Release 7.2) available at https://www.real-statistics.com/ (Purdue University, West Lafayette, IN, USA) (accessed on 15 April 2022).

## 3. Results

### 3.1. In Situ Analysis of hUC Histology and ECM Histochemistry

The in situ histological analysis of the hUC stained with HE confirmed the presence of and external layer composed of amnioblast cells, three inner vascular structures (an umbilical vein and two umbilical arteries), and a mucous connective tissue (Figure 1A). This mucous connective tissue can be divided in several zones with particular histological features. The first area is the intervascular zone (IV). This region is found in the central area of the hUC, forming a triangle-shaped structure among the blood vessels. As shown in Figure 1A, the ECM is well organized, and the cell density is relatively low. The second area is the perivascular zone (PV), which surrounds the blood vessel structures. Histologically, the PV zone shows a well-organized dense ECM containing abundant cells. The subamnioblastic zone (SAM) can be found below the superficial amnioblasts surrounding the hUC and is characterized by a very loose ECM and few cells. Finally, the Wharton zone (WH) can be identified below the SAM zone, in areas without close contact with the blood vessels. WH showed high cell density, and the ECM structure is less organized than in the IV and PV zones, but it is denser than in the SAM zone. Analysis of each zone using SEM confirmed that the area containing a denser ECM is the PV zone, whereas SAM had the looser ECM.

The analysis of relevant ECM components in the hUC sections using histochemistry revealed several differences among the hUC regions (Figure 1B and Table 1). First, AB analysis showed that the hUC contains high amounts of proteoglycans, but significant differences were found among the different areas. Specifically, the highest intensity of proteoglycans corresponded to the IV area (88.44 ± 28.69 I.U.), followed by WH (83.37±24.84 I.U.), PV (82.08 ± 24.38 I.U.), and SAM (76.08 ± 23.83 I.U.). Differences between SAM and IV were statistically significant. Then, the quantification of glycosaminoglycans as determined by PAS showed that the highest staining intensity was found in the PV zone (69.89 ± 32.66 I.U.), followed by IV (58.96 ± 30.41 I.U.), WH (55.88 ± 26.77 I.U.), and SAM (52.98 ± 29.03 I.U.). Differences between PV and the rest of zones were statistically significant. Finally, we analyzed the presence of collagen fibers in each area of the hUC using PR. Results showed that the highest intensity corresponded to PV (125.89 ± 30.24 I.U.) and IV (123.79 ± 26.74 I.U.), followed by WH (99.06 ± 21.76 I.U.) and SAM (91.44 ± 27.74 I.U.). Differences among samples were statistically significant, except for the comparison of IV vs. PV. Statistical p values for each comparison are shown in Table 1.

### 3.2. Immunohistochemical Analysis of MSC Markers in hUC and Isolated hUC-MSC

The characterization of relevant MSC markers in hUC cells was determined by immunohistochemistry for CD73, CD90, and CD105 markers. In the first place, the analysis in situ of hUC sections (Figure 2 and Table 1) revealed that CD73 was highly positive in all cells found in the four hUC zones (IV, PV, SAM, and WH), with no differences among zones. The CD90 marker revealed that the SAM region had the highest percentage of CD90-positive cells (73.97 ± 22.14%), followed by WH (36.95 ± 33.20%), IV (33.40 ± 35.41%), and PV (14.42 ± 20.78%). Differences were statistically significant for all comparisons, except for IV vs. WH. In addition, we found that the percentage of cells showing positive expression of the CD105 marker was very low in all zones (4.18 ± 11.41% in IV, 6.15 ± 16.53% in PV, 1.79 ± 8.73% in SAM, and 3.86 ± 9.69% in WH), with differences among zones being non-significant.

Next, we evaluated the expression of the same MSC markers in the hUC-MSC isolated from the cord and kept in ex vivo culture using flow cytometry. The results of this analysis (Figure 3) showed that cultured cells expressed high amounts of CD73 (an average 73.61% for all cell types), CD90 (85.46%), and CD105 (72.94%). Specific percentages of positive cells corresponding to each region are shown in Figure 3.

### 3.3. Immunohistochemical Analysis of Vimentin and ECM Components in hUC and Isolated hUC-MSC

On the one hand, in situ analysis of VIM expression, an intermediate filament protein expressed by mesenchymal stem cells, showed that all cells found at the four hUC zones (IV, PV, SAM, WH) were positive for this marker (Figure 4). On the other hand, we quantified several fibrillar and non-fibrillar components of the ECM using immunohistochemistry (Figure 5 and Table 1). The results first showed that collagen type III was especially abundant in PV (61.39 ± 14.08 I.U.) and WH (59.95 ± 11.80 I.U.), followed by SAM (54.69 ± 17.41 I.U.), and WH (53.32 ± 13.38 I.U.). Differences were significant for the comparison of IV vs. PV, IV vs. WH, and PV vs. SAM. All samples were also rich in collagen type IV (81.8 ± 10.49 I.U.) in PV, (76.35 ± 13.46 I.U.) in IV, (70.99 ± 15.59 I.U.) in WH, and (64.71 ± 19.33 I.U.) in SAM). All zones were statistically different from the rest of the zones. Then, we analyzed fibronectin and versican, as key non-fibrillar components of the ECM. For fibronectin, the highest values were found in PV (82.12 ± 6.49 I.U.), with high values also found in IV (79.63 ± 12.88 I.U.), WH (73.76 ± 15.60 I.U.), and SAM (69.48 ± 32.11 I.U.). Differences were statistically significant for the comparison of IV vs. WH, PV vs. SAM, and PV vs. WH. Finally, the highest contents of versican corresponded to SAM (53.92 ± 8.96 I.U.) and PV (52.88 ± 9.76 I.U.), followed by WH (49.69 ± 7.67 I.U.) and IV (45.23 ± 7.13 I.U.). Interestingly, hUC-MSC isolated from each area retained the same profile of the cells allocated in situ and showed high expressions of vimentin, collagens III and IV, and fibronectin, with lower expression of versican; no differences were detected among hUC-MSC isolated from different hUC zones (Table 2).

### 3.4. Immunohistochemical Analysis of Vascular Cell Markers in hUC and Isolated hUC-MSC

The analysis of three vascular markers showed that hUC cells intrinsically expressed some of these markers in situ in the hUC (Figure 5 and Table 1). For CD31, we found that the expression was only positive in PV, although at very low levels (8.03 ± 17.34 I.U.), with statistical differences from the other zones. Then, the analysis of CD34 revealed high expression of this cell marker in all samples, especially in WH (67.86 ± 136.18 I.U.), followed by SAM (53.05 ± 36.53 I.U.), IV (43.93 ± 38.43 I.U.), and PV (34.3 ± 30.62 I.U.). Differences were significant for the comparison of PV vs. SAM and PV vs. WH. Finally, we found that vWF expression was highly positive in all hUC regions, with the highest intensity found in IV (83.52 ± 4.84 I.U.), which was significantly higher than PV (78.19 ± 6.57 I.U.), SAM (78.38 ± 13.6 I.U.), and WH (77.88 ± 6.37 I.U.). When the isolated hUC-MSC were analyzed, we found a slightly positive CD31 signal in cells isolated from the PV region, but not from the rest of the hUC areas. For CD34, cultured hUC-MSC showed a positive signal only for cells isolated from the IV and PV areas, whereas vWF was strongly positive in hUC-MSC cultured from IV, positive in cells isolated from WH, and mildly positive in cells cultured from the PV and SAM areas (Figure 5 and Table 2).

## 4. Discussion

Although it is well known that the hUC contains diverse zones with substantial histological differences in their ECM matrix and cell profile [1,5,6], the specific potential of each zone as a source of cells for use in vascular tissue engineering has not been established to the date. In the present work, we first confirmed the existence of histological differences between the IV, PV, SAM, and WH zones. In fact, our in situ analysis showed that IV and PV were rich in collagen fibers, proteoglycans, and glycosaminoglycans, as determined by histochemistry, whereas SAM and WH tended to show lower concentrations of these ECM components. Although further biomechanical studies are needed, these findings suggest that the biomechanical properties of the hUC could vary among the specific zones. As the local microenvironment provided by fibrillar and non-fibrillar components of the ECM allows cells to communicate and regulate cell fate and behavior, we may hypothesize that the specific properties of each zone could be related to their particular physical properties.

These results could be related to the proximity of IV and PV to the umbilical blood vessels and the high pressures that must be withstood by these structures [20,21]. Furthermore, the fact that PV showed the highest amount of glycosaminoglycans could be explained by the existence of a basement membrane surrounding each blood vessel in this area. In general, our findings confirm the presence of dense areas (mainly consisting of PV and IV) and loose areas (mainly made up of SAM and WH) in the hUC and confirm the heterogeneity of the different regions of the hUC [1,5,6].

These structural differences detected by histology and histochemical assays led us to study the stemness profile of cells allocated at each hUC region. It has been previously established that cells used in tissue engineering and cell therapy should meet several criteria established by the International Society for Cellular Therapy (ISCT) for cells [22]. These include, among others, the expression of several surface markers such as CD73, CD90, and CD105 and the lack of expression of CD14 (or CD11b), CD34, CD45, CD79a (or CD19), and human leukocyte antigen (HLA)-DR molecules [22]. Expression of these markers has been described as necessary for the cells to properly exert their pro-regenerative functions both ex vivo and in vivo. In this regard, we found that all cells allocated in the four hUC regions were positive for CD73, whereas a low percentage of cells were positive for CD105, with no differences among zones. However, the in situ analysis found significant differences for CD90, with high values in SAM as compared to the other regions. Interestingly, the flow cytometry analyses carried out on hUC-MSC isolated from each region and kept in culture revealed highly positive expression of these three markers in the four types of cells, with few differences among areas. The fact that all isolated hUC-MSC kept ex vivo expressed these markers is intriguing and suggests that these cells could be activated by the culture conditions once isolated from the hUC, thus confirming the putative utility of hUC-MSC for general use in tissue engineering independent of their specific allocation in the hUC. However, a trend was found to a higher expression of CD73 and CD105 in cells isolated from SAM. Although future studies should confirm this finding, previous reports already suggested that SAM cells could be especially useful for regenerative medicine applications [23,24]. Whether this finding is associated with a different differentiation potential of each hUC zone remains to be determined.

On the other hand, we carried out several immunohistochemical analyses to identify several key components of the hUC. In the first place, we confirmed that all cells were positive for vimentin, both in situ and after ex vivo isolation. As a stromal marker playing an important role in maintaining cell integrity, the presence of vimentin confirms the stromal nature of these cells [25,26]. In the second place, we analyzed several ECM components, including COL-III, COL-IV, FBN, and VERS. Results demonstrated that all cells in the hUC had the potential to synthetize fibrillar and non-fibrillar components of the ECM. In fact, PV showed the highest levels of COL-III, COL-IV, and FBN, which is in agreement with the results found for the histochemical analyses and could explain the high density of this specific region of the hUC. Strikingly, all hUC-MSC isolated from the different areas and kept ex vivo were able to express all these markers by immunohistochemistry, suggesting again that cultured cells keep their profile once transferred to the culture conditions. Again, the absence of differences among cells kept ex vivo suggests that cells are able to partially modify their phenotype once isolated and cultured, independently of their specific origin in the hUC.

Finally, we analyzed three key markers of vascular cell differentiation in the hUC cells to determine the potential usefulness of these cells in vascular tissue engineering. The identification of an adequate source of cells with vascular potential could be very useful for the generation of large vascular structures resembling arteries and veins [27,28], but also to induce the rapid formation of a microvascular network in skin, oral mucosa, and other bioengineered tissues, as previously suggested [29,30]. 

In this regard, we found that CD31, a protein with an important role in endothelial cell adhesion [31], was negative in most hUC regions, but showed a certain signal in PV, probably due to the more differentiated profile of these cells attributed to their low expression of CD90 and increased amount of ECM components. Although this in situ expression was low, it could be explained by the proximity of this region to the hUC blood vessels. Interestingly, this slightly positive expression was also found in isolated hUC-MSC, suggesting that this profile is kept after isolation and ex vivo culture. Moreover, we found that a high percentage of cells showed positive expression of CD34 in situ. The intrinsic expression of this vascular marker also suggests that hUC cells have the potential to differentiate to vascular cells, since CD34 is known to be highly expressed by endothelial progenitor cells committed to develop small blood vessels, especially in the case of the small capillaries [32]. However, isolation and transfer to the ex vivo culture conditions resulted in a positive expression of CD34 in cells isolated from IV and PV, but cells isolated from SAM and WH were negative for this marker. Although these results will require future study, we may hypothesize that IV and PV cells, which are found in a region that is close to the blood vessels, could have increased potential for use in vascular tissue engineering once isolated and cultured ex vivo. In the third place, we found a high expression of vWF markers by cells corresponding to all hUC regions, especially in IV cells. Found in the Weibel–Palade bodies of endothelial cells, vWF is a key marker of these cells and plays a role in cell adhesion and vascular cell homeostasis [33]. The finding that all regions were highly positive for this marker once again supports the potential of hUC cells to differentiate towards the vascular cell type lineage. In addition, we found that isolated hUC-MSC retained vWF expression ex vivo, especially in the case of the cells isolated from IV, as expected from these cells allocated close to the vascular structures in the hUC. Altogether, these results confirm the potential of hUC cells to differentiate to the vascular cell lineage, as previously suggested [34]. Since all cell types were able to express characteristic vascular markers, we might hypothesize that all cell types analyzed in the present work could have a common embryological origin that could be related to embryonic allantoid vessels.

The present study has some drawbacks and limitations that should be addressed in future research. First, independent studies should evaluate the multilineage mesodermal differentiation potential of each hUC zone. Second, the transitional zones allocated among the different hUC zones should be studied to clarify the differences between hUC zones. Furthermore, long-term studies should be focused on the identification of changes in the expression of vascular markers in subsequent subcultures of cells derived from the four hUC zones to determine the most suitable cell passage for tissue engineering applications. Finally, our results were obtained using a small group of samples. Therefore, results should be validated in a larger cohort of cords to assess intersample variability.

In summary, our results confirm the heterogeneity of the different regions of the hUC and the need for selecting specific regions for tissue engineering purposes. Although cells showed different profiles in situ, we found that the isolation and culture of hUC-MSC corresponding to different regions resulted in a very similar profile of cultured cells for some parameters, but specific differences were found for the vascular markers CD31, CD34, and vWF, suggesting that cells corresponding to IV should be preferentially isolated for vascular tissue engineering. In addition, the high potential of cells isolated from the umbilical cord suggests that these cells could also be used for the generation of other human organs and tissues such as the human skin, cornea, and oral mucosa. 

## Figures and Tables

**Figure 1 jpm-12-00648-f001:**
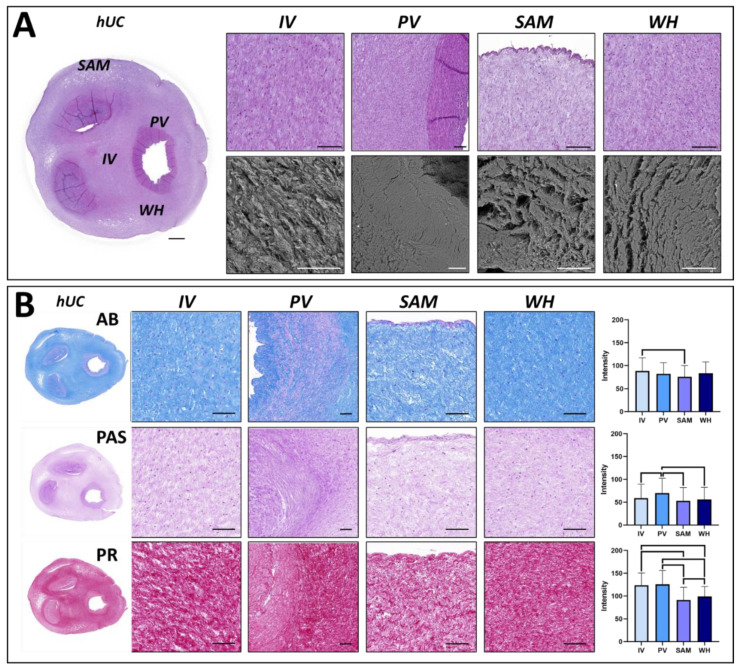
In situ histological (panel (**A**)) and histochemical analysis (panel (**B**)) of the human umbilical cord (hUC) and its different zones. Samples corresponding to the histological analysis were stained with hematoxylin-eosin (HE) or analyzed by scanning electron microscopy (SEM). Samples analyzed by histochemistry were subjected to alcian blue (AB), periodic acid-Schiff (PAS), and picrosirius red (PR) staining. The histochemical intensity obtained for each sample and each method is shown in the histograms to the right, with statistically significant differences shown as brackets. IV: intervascular zone; PV: perivascular zone; SAM: subamnioblastic zone; WH: Wharton zone. Scale bar = 200 µm.

**Figure 2 jpm-12-00648-f002:**
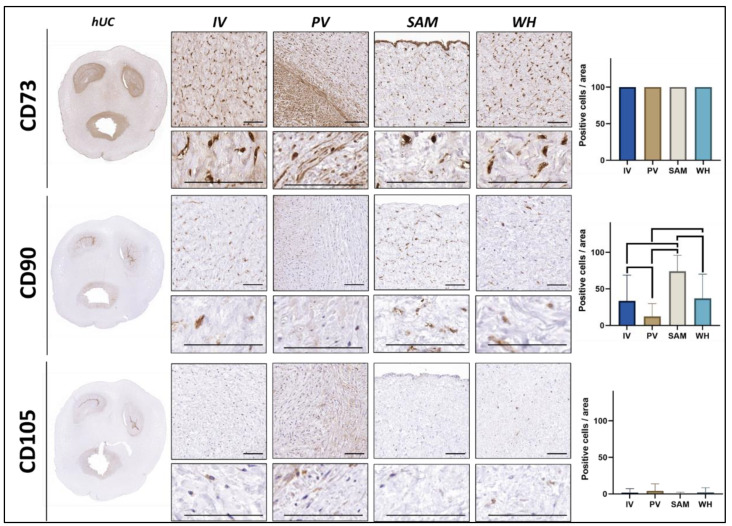
In situ immunohistochemical analysis of relevant MSC markers in cells found in the human umbilical cord (hUC) and its different zones. Images obtained at two different magnifications are shown for each marker. Quantification of the percentage of positive cells found in each sample and each method is shown in the histograms to the right, with statistically significant differences shown as brackets. IV: intervascular zone; PV: perivascular zone; SAM: subamnioblastic zone; WH: Wharton zone. Scale bar = 200 µm.

**Figure 3 jpm-12-00648-f003:**
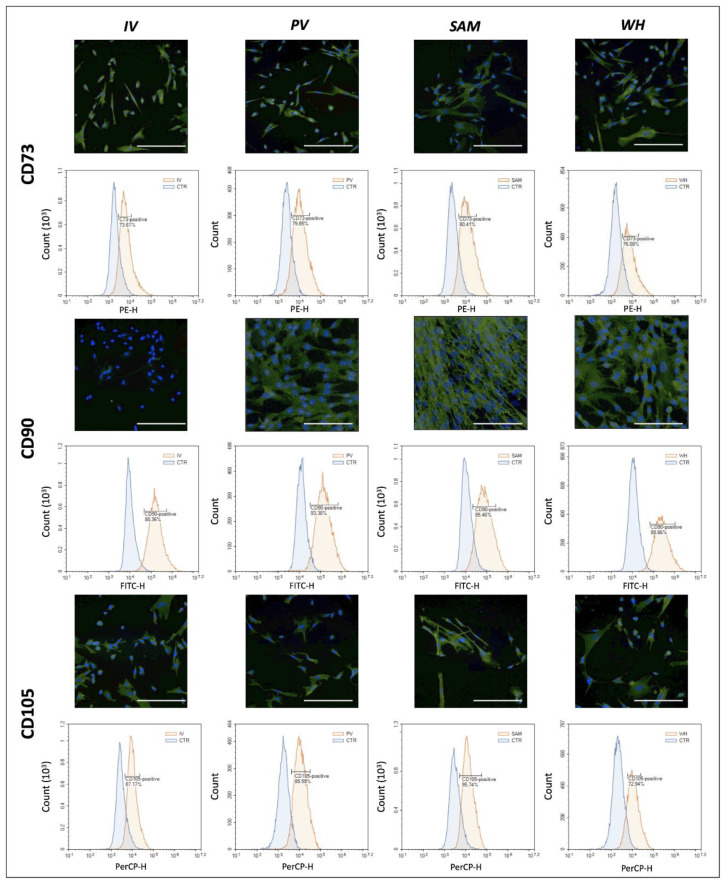
Analysis of relevant MSC markers in cells isolated from each zone of the human umbilical cord (hUC-MSC). For each marker, immunofluorescence images and flow cytometry distributions are shown. In the immunofluorescence images, a positive signal is shown in green, whereas cell nuclei are shown in blue. Scale bar = 200 µm.

**Figure 4 jpm-12-00648-f004:**
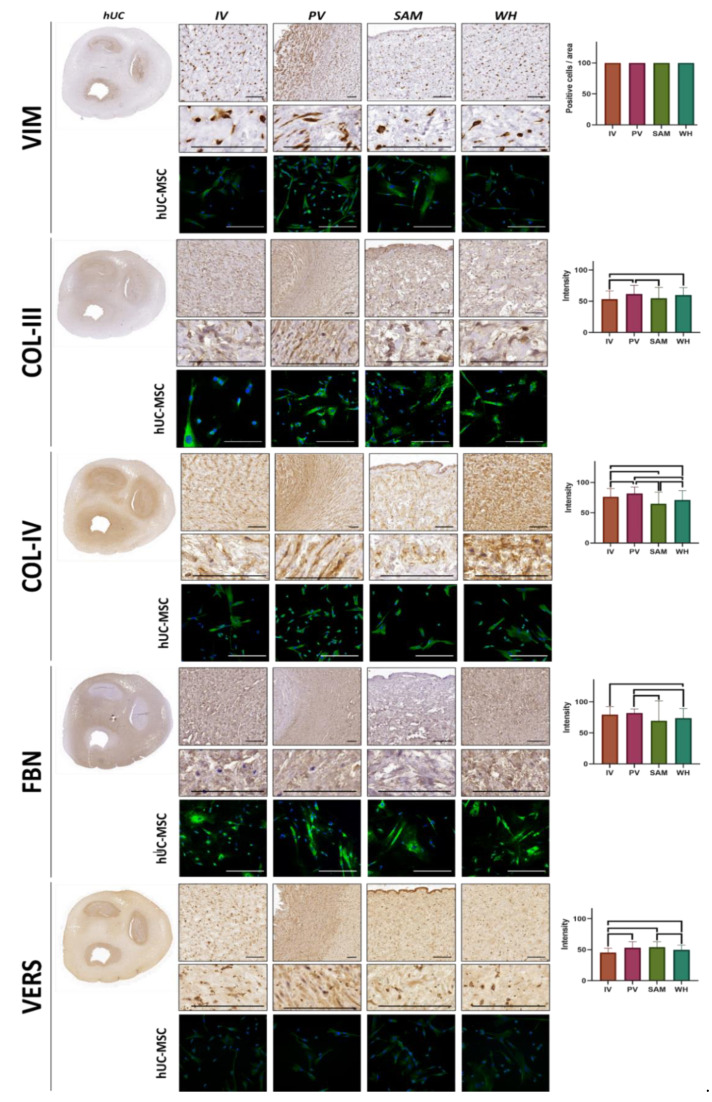
Immunohistochemical analysis of vimentin and four fibrillar (collagen type III—COL-III, and collagen type IV—COL-IV) and non-fibrillar (fibronectin—FBN, and versican—VERS) components of the ECM in the human umbilical cord (hUC) and its different zones. Images obtained at two different magnifications are shown for each marker. Quantification of the percentage of positive cells found in each sample and each method is shown in the histograms to the right, with statistically significant differences shown as brackets. Immunofluorescence images correspond to cells isolated from the hUC and kept in ex vivo culture (hUC-MSC), with positive signals shown in green, and cell nuclei shown in blue. IV: intervascular zone; PV: perivascular zone; SAM: subamnioblastic zone; WH: Wharton zone. Scale bar = 200 µm.

**Figure 5 jpm-12-00648-f005:**
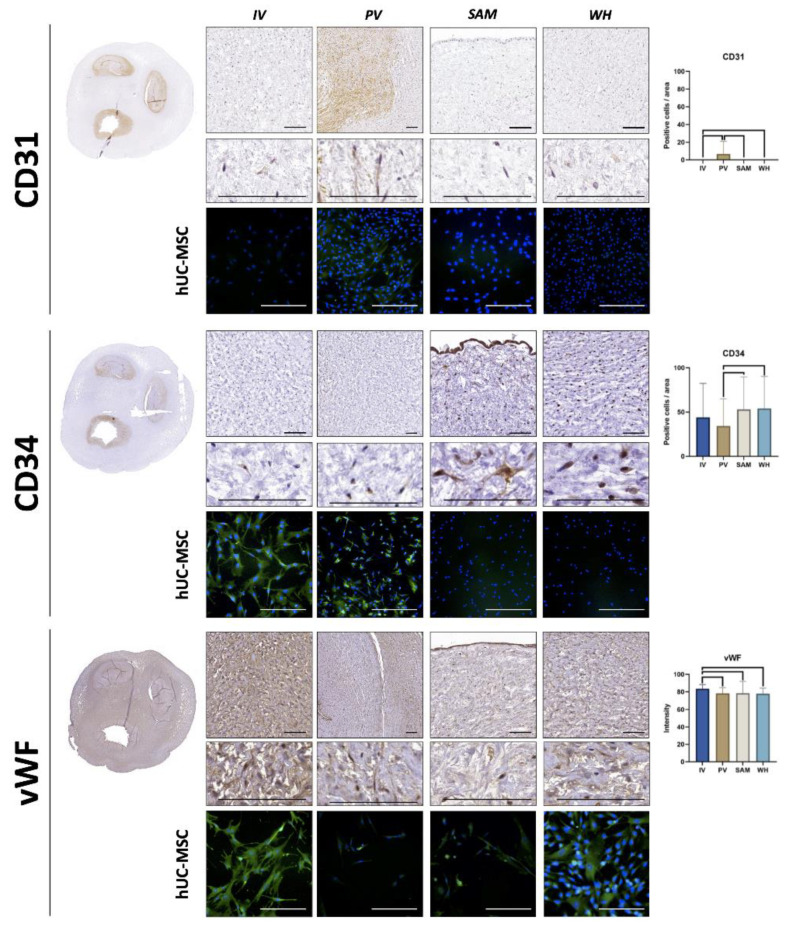
Immunohistochemical analysis of three vascular markers (CD31, CD34, and von Willebrand factor—vWF) in the human umbilical cord (hUC) and its different zones. Images obtained at two different magnifications are shown for each marker. Quantification of the percentage of positive cells found in each sample and each method is shown in the histograms to the right, with statistically significant differences shown as brackets. Immunofluorescence images correspond to cells isolated from the hUC and kept in ex vivo culture (hUC-MSC), with positive signals shown in green, and cell nuclei shown in blue. IV: intervascular zone; PV: perivascular zone; SAM: subamnioblastic zone; WH: Wharton zone. Scale bar = 200 µm.

**Table 1 jpm-12-00648-t001:** Quantitative analysis of the markers analyzed in the present work by histochemistry and immunohistochemistry in each zone of the human umbilical cord (hUC). Average and standard deviations are shown for each zone, and the statistical *p* values corresponding to the comparison between two specific zones are shown in the columns to the right. IV: intervascular zone; PV: perivascular zone; SAM: subamnioblastic zone; WH: Wharton zone; AB: alcian blue; PAS: periodic acid-Schiff; PR: picrosirius red; VIM: vimentin; COL-III: collagen type III; COL-IV: collagen type IV; FBN: fibronectin; VERS: versican; vWF: von Willebrand factor. For AB, PAS, PR, COL-III, COL-IV, FBN, VERS, and vWF, values correspond to staining intensity units (I.U.), whereas the results shown for CD73, CD90, CD105, VIM, CD31, and CD34 correspond to the percentage of positive cells.

	IV	PV	SAM	WH	IV vs. PV	IV vs. SAM	IV vs. WH	PV vs. SAM	PV vs. WH	SAM vs. WH
AB	88.44 ± 28.69	82.08 ± 24.38	76.08 ± 23.83	83.37 ± 24.84	0.1330	0.0012	0.0822	0.0974	0.5827	0.0754
PAS	58.96 ± 30.41	69.89 ± 32.66	52.98 ± 29.03	55.88 ± 26.77	0.0035	0.0837	0.5050	<0.0001	0.0013	0.3365
PR	123.79 ± 26.74	125.89 ± 30.24	91.44 ± 27.74	99.06 ± 21.76	0.3616	<0.0001	<0.0001	<0.0001	<0.0001	0.0296
CD73	100 ± 0	100 ± 0	100 ± 0	100 ± 0	1.0000	1.0000	1.0000	1.0000	1.0000	1.0000
CD90	33.4 ± 35.41	14.42 ± 20.78	73.97 ± 22.14	36.95 ± 33.2	0.0010	<0.0001	0.4978	<0.0001	<0.0001	<0.0001
CD105	4.18 ± 11.41	6.15 ± 16.53	1.79 ± 8.73	3.86 ± 9.69	0.5498	0.3032	0.9238	0.0963	0.6164	0.2514
VIM	100 ± 0	100 ± 0	100 ± 0	100 ± 0	1.0000	1.0000	1.0000	1.0000	1.0000	1.0000
COL-III	53.32 ± 13.38	61.39 ± 14.08	54.69 ± 17.41	59.95 ± 11.8	0.0001	0.5594	0.0064	0.0074	0.0721	0.0546
COL-IV	76.35 ± 13.46	81.8 ± 10.49	64.71 ± 19.33	70.99 ± 15.59	0.0332	<0.0001	0.0238	<0.0001	<0.0001	0.0127
FBN	79.63 ± 12.88	82.12 ± 6.49	69.48 ± 32.11	73.76 ± 15.6	0.8967	0.0564	0.0059	0.0177	0.0016	0.2123
VERS	45.23 ± 7.13	52.88 ± 9.76	53.92 ± 8.96	49.69 ± 7.67	<0.0001	<0.0001	0.0001	0.0744	0.0997	0.0007
CD31	0 ± 0	8.03 ± 17.34	0 ± 0	0 ± 0	0.0203	0.9989	0.9989	0.0203	0.0203	0.9989
CD34	43.93 ± 38.43	34.3 ± 30.62	53.05 ± 36.53	67.86 ± 136.18	0.1670	0.2683	0.0858	0.0012	0.0004	0.6613
vWF	83.52 ± 4.84	78.19 ± 6.57	78.38 ± 13.6	77.88 ± 6.37	<0.0001	0.0304	<0.0001	0.2965	0.6144	0.1806

**Table 2 jpm-12-00648-t002:** Semiquantitative analysis of vimentin, four fibrillar (collagen type III—COL-III, and collagen type IV—COL-IV), and non-fibrillar (fibronectin—FBN, and versican—VERS-) components of the ECM and three vascular markers (CD31, CD34, and von Willebrand factor—vWF) in cells isolated from each zone of the hUC and kept in ex vivo culture (hUC-MSC). IV: intervascular zone; PV: perivascular zone; SAM: subamnioblastic zone; WH: Wharton zone. Expression was classified as strongly positive (+++), positive (++), mildly positive (+), slightly positive (+/−), or negative (−).

	IV	PV	SAM	WH
VIM	++	++	++	++
COL-III	+++	+++	+++	+++
COL-IV	++	++	++	++
FBN	+++	+++	+++	+++
VERS	+	+	+	+
CD31	−	+/−	−	−
CD34	++	++	−	−
vWF	+++	+	+	++

## Data Availability

Data is contained within the article or Appendix A.

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
