# Peer review of "Histological Profiling of the Human Umbilical Cord: A Potential Alternative Cell Source in Tissue Engineering"

_jpm, 2022, doi:10.3390/jpm12040648_

Round 1
Reviewer 1 Report
The manuscript by Blanco-Elices and cols. aims to characterize the human umbilical cord regions and their potential use in vascular regeneration. Umbilical cords specimen were divided in four regions, namely intervascular, perivascular, subaminoblastic, and Wharton jelly. Histological analysis confirmed the heterogeneity of the different regions of the umbilical cord. Primary cell cultures of mesenchymal stem cells (hUC-MSC) were also isolated from each region were also isolated.
- Extracellular matrix composition and stuffiness of the microenvironment play a role in the differentiation of stem cells. Considering that the different regions of the umbilical cord contain different proportion of cells versus extracellular matrix how does this affect the potential use of cells for tissue engineering applications?
- Do the cells isolated from the different regions have the same myogenic, adipogenic, and osteogenic differentiation potential? Does a higher number of cells present in the perivascular and Wharton zone equates to higher stem cell potential?
- Within each zone what type of cell is the most predominant? What percentage of cells have acquired a mature phenotype?
- For the dissection how was the analysis of ECM components performed at “transition” zones?
- In the ex vivo culture studies, following tissue digestion was there any phenotype selection performed or all cells obtained were placed under culture? Interesting result showing the expression of CD105. Is this due to cell selection during culture or marker expression stimulation during cell culture?
- What characteristics of the cells located in PV, SAM and WH suggest that they are more suitable for human skin, cornea, and oral mucosa tissue engineering applications as stated in the final paragraph of the conclusion?
- The vWF expression pattern was very interesting, showing positive staining in all four regions. Can the authors propose a theory as to why this marker is present even in the non vascular regions?
Author Response
- Extracellular matrix composition and stuffiness of the microenvironment play a role in the differentiation of stem cells. Considering that the different regions of the umbilical cord contain different proportion of cells versus extracellular matrix how does this affect the potential use of cells for tissue engineering applications?
Author´s Response (AU): This is an interesting observation. We agree with the reviewer that the histological differences found among the different hUC zones for the ECM and cell content could influence their pro-regenerative functions. The fact that the content of fibrillar and non-fibrillar ECM components vary among the different zones could also imply that the biomechanical properties of each zone could be different. This interesting information has been included in the discussion section of the manuscript (Page 12, lines 346-351).
- Do the cells isolated from the different regions have the same myogenic, adipogenic, and osteogenic differentiation potential? Does a higher number of cells present in the perivascular and Wharton zone equates to higher stem cell potential?
(AU): Previous studies revealed that human umbilical cord derived-cells have the intrinsic potential to differentiate into myogenic, adipogenic and osteogenic linages (Alaminos et al., 2010). However, the potential of the different human umbilical cord zones (PV, IV, SAM and WH) to differentiate into mesoderm-like tissues have been unexplored to the date, and future works will have to address this issue. However, our preliminary results revealed that WH cells expressed high levels of the undifferentiation markers CD90 and CD73, whereas PV showed lower levels of CD90. Whether this differential expression of relevant markers is associated to a different differentiation potential will need to be evaluated in future research works. This information has been added in the revised version of the manuscript (Page 12, lines 380-381, Page 13, lines 432-433).
Alaminos, M., Pérez-Köhler, B., Garzón, I., García-Honduvilla, N., Romero, B., Campos, A., et al. (2010). Transdifferentiation potentiality of human Wharton’s jelly stem cells towards vascular endothelial cells. J. Cell. Physiol. 223, 640–647. doi:10.1002/jcp.22062.
- Within each zone what type of cell is the most predominant? What percentage of cells have acquired a mature phenotype?
(AU): In the present work, we found that all cell types were positive for CD90, CD105 and CD73, suggesting that all cells showed an undifferentiated profile. However, cells isolated form PV zone had the lowest values of CD90, suggesting that these cells could be more mature or more differentiated than the other cell types. This information is included in the discussion section (Page 13, lines 404-406).
- For the dissection how was the analysis of ECM components performed at “transition” zones?
(AU): In this work, we analyzed four umbilical cord regions. However, the transition zones were not analyzed, and will need to be characterized in independent works. This information has been included as a future research plan in the discussion section of the revised version (Page 13, lines 433-435).
- In the ex vivo culture studies, following tissue digestion was there any phenotype selection performed or all cells obtained were placed under culture? Interesting result showing the expression of CD105. Is this due to cell selection during culture or marker expression stimulation during cell culture?
(AU): For the ex vivo cell culture studies, small biopsies were obtained from each topographical zone of the human umbilical cord (PV, IV, SAM and WH) using a stereomicroscope. Biopsies corresponding to the different zones were enzymatically digested and harvested cells were cultured using standard cell culture conditions. No phenotype selection was performed on the isolated cells. We agree that the CD105 expression results are interesting. Although future works should determine the basis of this phenomenon, we could hypothesize that a this could be related to expression stimulation during cell culture. In fact, it was previously demonstrated that surface antigens vary according to the specific cell culture conditions used. The detailed isolation procedure has been described in the materials and methods section (Page 3, lines 101-103), and the expression of CD105 has been discussed in the manuscript (Page 12, lines 373-375).
- What characteristics of the cells located in PV, SAM and WH suggest that they are more suitable for human skin, cornea, and oral mucosa tissue engineering applications as stated in the final paragraph of the conclusion?
(AU): Previous works developed by our research group have demonstrated the putative usefulness of these cells for use in tissue engineering, and their pluripotential differentiation capability. Specifically, cells derived from WH zone were able to differentiate into non-mesodermal linages, including the epithelial lineage, and were able to differentiate into cells of the skin, oral mucosa and cornea. This information is included in the introduction and discussion sections of the manuscript (Page 2, line 53-55, Page 14, lines 446-448).
- The vWF expression pattern was very interesting, showing positive staining in all four regions. Can the authors propose a theory as to why this marker is present even in the non-vascular regions?
(AU): We could hypothesize that all the cells analyzed here could have a common embryological origin, and that the precursor cell could be derived from allantoid vessels. This hypothesis has been included in the discussion section of the revised manuscript (Page 13, lines 428-430).
Reviewer 2 Report
This is a well-written article on the histological and immunohistochemical particularities of different regions of the human umbilical cord. The description of the different portions of the hUC in situ corresponds to what has already been reported in the literature and does not add any innovation. The greatest merit of the article is the comparison of immunohistochemical staining between tissue in situ and ex vivo, showing the effect of culture in cell metabolism. In particular, the results with vascular cell markers add an innovation to the knowledge in the area.
The experimental design is adequate and the images are of excellent quality.
Although the results presented are not very innovative, it is a work carried out with responsibility and competence.
Author Response
This is a well-written article on the histological and immunohistochemical particularities of different regions of the human umbilical cord. The description of the different portions of the hUC in situ corresponds to what has already been reported in the literature and does not add any innovation. The greatest merit of the article is the comparison of immunohistochemical staining between tissue in situ and ex vivo, showing the effect of culture in cell metabolism. In particular, the results with vascular cell markers add an innovation to the knowledge in the area.
The experimental design is adequate and the images are of excellent quality.
Although the results presented are not very innovative, it is a work carried out with responsibility and competence.
Authors' response: Thank you for your kind comments. The results showed in this manuscript could open a new research line in the field of tissue engineering with future studies with more innovative approaches.
Reviewer 3 Report
The manuscript of Blanco-Elices et al., entitled ‘Histological profiling of the human umbilical cord: A potential alternative cell source in tissue engineering’ is devoted to studying the histological, histochemical and immunohistochemical analysis of the human umbilical cord (hUC) cells and extracellular matrix in order to evaluate the profile of different four main regions of the hUC: the superficial subamnioblastic zone (SAM), the perivascular zone (PV), the Wharton zone (WH) between SAM and PV, and the intervascular zone (IV) in the central among the three vessels of the hUC.
This is a well designed study and could be accepted for publication in the Journal of Personalized Medicine after a few minor changes.
My comments and suggestions are:
- Material and Methods. At what passage were cultured hUCs used for surface marker analysis? Is there a difference after passage 1-2-3? Have you seen any changes in the passage process? Did the vascular markers CD31, CD34, and vWF, disappear during the next passages? Please specify this in the Materials and Methods section.
- The font of the caption to Fig. 3 (next to the diagrams) is small, so that it can be read well, the font should be enlarged.
- Please, explain (or correct) the inconsistencies in Fig. 5. Discrepancy between diagrams and cell pictures. For example, CD34. On the left in the pictures, cells are visualized a lot in IV and PV regions, but on the right in the diagrams, on the contrary, more on the right, there are more cells in other zones (SAM and WH). Explain this or provide more appropriate illustrations.
- Can you confidently state that the results obtained after testing 5 hUCs will be identical to the results obtained after testing a much larger number of hUCs from donors of different age, somatic and genetic status?
- A total of 5 umbilical cords were investigated. For a laboratory study, it may be sufficient to draw reliable conclusions. However, the amount of hUC can be increased in the future. It is well known that primary cell cultures properties are quite varied from donor to donor. Therefore, it is not yet possible to state with certainty whether the conclusions drawn depend on race, gender, the course of pregnancy, the pathology of the mother, her age ... etc. At the moment, we cannot say with certainty that the results obtained after testing a much larger number of the hUCs will be identical to the results obtained in this article.
- We all understand that there are no comprehensive studies that answer all scientific questions. It would be useful for the authors to cite the weaknesses of the work done at the end of the manuscript, thus outlining a plan for future research. For example, weaknesses and prospects for your work are an increase in the number of umbilical cords, passages ... etc. To confirm the scientific hypothesis and further personalized application of the results in the experiment in practice, it is necessary to continue research in this direction. Perhaps you know that your future research on the creation of vascular prostheses will be interesting. Further experimental studies in vivo will be very important.
- The manuscript contains a fairly complete selection of literature, covering both early work on the identification of progenitor cells and recent work. In the manuscript, 6 out of 34 references are references to the works of one or more authors (Nos. 6, 12, 13, 14, 19, 29, and 34). However, these references are indicated in the correct places in the manuscript, demonstrating the logical continuation of research by the University scientific team. I would like to hope to see the results of your research in the future not only in situ and ex vivo, but also with the transition to in vivo experiments on animals and to the clinic.
Author Response
- Material and Methods. At what passage were cultured hUCs used for surface marker analysis? Is there a difference after passage 1-2-3? Have you seen any changes in the passage process? Did the vascular markers CD31, CD34, and vWF, disappear during the next passages? Please specify this in the Materials and Methods section.
Author´s Response (AU): This is an interesting question. All cells used in the present work correspond to early passages (1 to 3) to prevent the cells from losing their differentiation status. As the reviewer states, cells tend to lose their potential when sequentially passaged in culture, as we previously demonstrated (Martin-Piedra et al., 2014). This important issue has been included in the discussion section of the revised version of the manuscript (Page 13, lines 435-437).
Martin-Piedra, M. A., Garzon, I., Oliveira, A. C., Alfonso-Rodriguez, C. A., Carriel, V., Scionti, G., et al. (2014). Cell viability and proliferation capability of long-term human dental pulp stem cell cultures. Cytotherapy 16, 266–277. doi:10.1016/j.jcyt.2013.10.016.
- The font of the caption to Fig. 3 (next to the diagrams) is small, so that it can be read well, the font should be enlarged.
(AU): Figure 3 have been improved for clarity, as suggested.
- Please, explain (or correct) the inconsistencies in Fig. 5. Discrepancy between diagrams and cell pictures. For example, CD34. On the left in the pictures, cells are visualized a lot in IV and PV regions, but on the right in the diagrams, on the contrary, more on the right, there are more cells in other zones (SAM and WH). Explain this or provide more appropriate illustrations.
(AU): We appreciate this observation. The diagram to the right corresponds to the average of several samples analyzed in this work, whereas the figures to the left are just illustrative images. Following the reviewer’s suggestion, we have replaced the images by more representative images of the average results.
- Can you confidently state that the results obtained after testing 5 hUCs will be identical to the results obtained after testing a much larger number of hUCs from donors of different age, somatic and genetic status?
(AU): In this work, we have analyzed 5 human umbilical cords, and the results were very homogeneous among the 5 samples. However, it is clear that results would have higher scientific value and statistical power if the sample size could be increased and if the results could be confirmed in a larger cohort of samples. This important statement has been included as limitation of the study (Page 13, lines 438-439).
- A total of 5 umbilical cords were investigated. For a laboratory study, it may be sufficient to draw reliable conclusions. However, the amount of hUC can be increased in the future. It is well known that primary cell cultures properties are quite varied from donor to donor. Therefore, it is not yet possible to state with certainty whether the conclusions drawn depend on race, gender, the course of pregnancy, the pathology of the mother, her age ... etc. At the moment, we cannot say with certainty that the results obtained after testing a much larger number of the hUCs will be identical to the results obtained in this article.
(AU): As stated above, we agree that the number of samples should be increased, and that the use of small sample size could be influenced by intersample variability. This important statement has been included as limitation of the study (Page 13, lines 438-439).
- We all understand that there are no comprehensive studies that answer all scientific questions. It would be useful for the authors to cite the weaknesses of the work done at the end of the manuscript, thus outlining a plan for future research. For example, weaknesses and prospects for your work are an increase in the number of umbilical cords, passages ... etc. To confirm the scientific hypothesis and further personalized application of the results in the experiment in practice, it is necessary to continue research in this direction. Perhaps you know that your future research on the creation of vascular prostheses will be interesting. Further experimental studies in vivo will be very important.
(AU): We appreciate the valuable comments of the reviewer. In the revised version of the manuscript, we have included the limitations of the study, and we suggest several future experiments that should be carried out to complement the present work (Page 12, lines 346-348 and 378-381, Page 13, lines 431-439).
- The manuscript contains a fairly complete selection of literature, covering both early work on the identification of progenitor cells and recent work. In the manuscript, 6 out of 34 references are references to the works of one or more authors (Nos. 6, 12, 13, 14, 19, 29, and 34). However, these references are indicated in the correct places in the manuscript, demonstrating the logical continuation of research by the University scientific team. I would like to hope to see the results of your research in the future not only in situ and ex vivo, but also with the transition to in vivo experiments on animals and to the clinic.
(AU): Thank you very much for your valuable comments.
Round 2
Reviewer 1 Report
This reviewer would like to thank the authors for their careful and insightful responses and for taking in consideration the suggestions and modifying the manuscript accordingly.